# Geometric Projection of Information Manifolds for Robust Decision-Making with LLMs in Adversarial Driving Environments

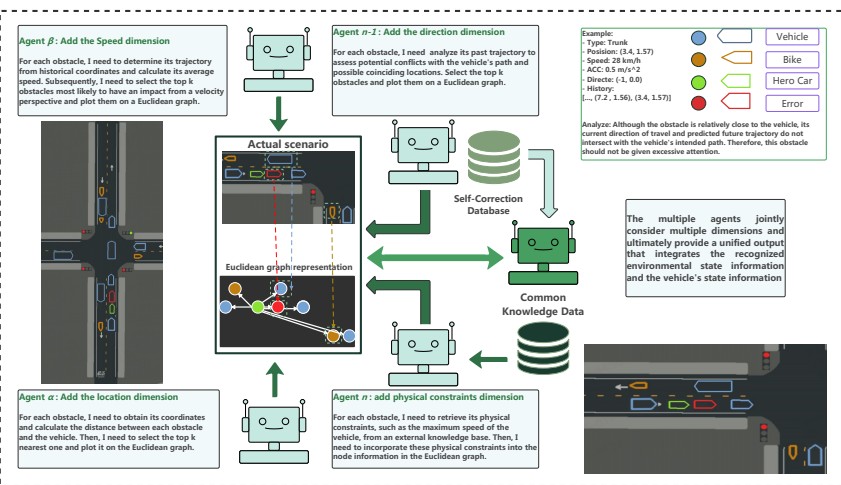

Figure 1: Overall framework of our manifold-enhanced LLM approach for autonomous driving with multiple agents (1 to n) that decouple information from various perspectives to form a decision-making manifold.

## Abstract

Perception uncertainty poses a critical challenge for autonomous driving systems (ADS), where small-probability anomalies can lead to catastrophic failures in decision-making. While existing approaches rely on redundant sensors or multi-modal fusion, they struggle with rare edge cases and require extensive datasets for training. We propose LLM-ADF, a Large Language Model-based Autonomous Driving Framework that leverages few-shot learning to enhance robustness against perceptual anomalies. Our key innovation lies in constructing a specialized autonomous driving space through information geometry-guided dimensionality reduction, decoupling high-dimensional text embeddings into driving-relevant features while preserving contextual reasoning capabilities. We introduce a manifold-based reasoning mechanism that connects the text space with the driving space, enabling LLMs to perform spatial-temporal inference even under corrupted inputs. The framework incorporates a self-correction database that enables continuous learning from historical anomalies, dynamically adjusting the manifold structure through Fisher information metrics. We construct an adversarial dataset with 2,730 anomalous frames simulating sensor failures and adversarial attacks. Experimental results on UniAD and ST-P3 benchmarks demonstrate that LLM-ADF achieves 24.93% average collision rate on UniAD, outperforming GPT-Driver by 22% under normal conditions and showing 14.9% degradation under anomalies compared to 17-21% for existing LLM-based methods. Our approach represents a paradigm shift towards few-shot learning in safety-critical autonomous systems, providing theoretical foundations and practical solutions for L4 autonomous driving deployment.

# 1 INTRODUCTION

Perception uncertainty remains a critical challenge for Autonomous Driving Systems (ADS). Despite existing approaches like redundant perception systems and multi-modal fusion techniques (Shao et al., 2024; Feng et al., 2024), significant deficiencies persist when handling low-probability perceptual anomalies. Though rare, these anomalies can introduce major safety hazards to decision modules, directly threatening the overall system safety (He & Lv, 2023). The ongoing coordination issues between perception and decision modules further exacerbate this problem (Kim et al., 2015), as there are no effective mechanisms for communicating uncertainty.

Deep learning models in autonomous driving typically rely on large-scale datasets and computational resources (Chen et al., 2023), presenting significant challenges in the context of globally diverse traffic environments and rapid iterative development needs (Huang et al., 2022). The scarcity of anomalous scenario data further constrains model performance (Xue et al., 2024), particularly in autonomous driving contexts requiring real-time responses. Consequently, few-shot learning for anomalous inputs has emerged as an important paradigm to reduce data dependencies and improve system generalization capabilities (Song et al., 2023), representing an inevitable trend in intelligent transportation system development (Li & Huang, 2022; Hong et al., 2024).

However, applying few-shot learning in ADS faces dual challenges (Shen et al., 2022). On one hand, training and inference with limited samples require models to not only understand complex spatial environments (Sural et al., 2024) but also extract key environmental features (Li & Shi, 2022). On the other hand, such learning approaches exhibit high sensitivity to input anomalies, while existing systems lack effective mechanisms for transmitting perception uncertainties to decision endpoints (Tang et al., 2022), and decision modules typically lack specialized training for anomalous scenarios. Therefore, designing decision systems that maintain inference stability under anomalous inputs is crucial for enhancing ADS robustness (Rafique et al., 2024).

The emergence of Large Language Models (LLMs) offers a new approach to addressing these challenges. The extensive driving-related knowledge acquired during pre-training and strong contextual reasoning capabilities provide a foundation for decision-making in anomalous situations (Yang et al., 2023b). However, current LLMs lack specialized training for autonomous driving scenarios, limiting their direct application (Ma et al., 2025). Consequently, effectively integrating LLMs' cognitive advantages with the specialized requirements of autonomous driving systems becomes an urgent scientific problem (Chen & Lu, 2024).

Based on these considerations, this research proposes an innovative LLM-based autonomous driving framework specifically optimized for few-shot learning and anomalous input processing. We construct a semantically enhanced autonomous driving space, achieving feature disentanglement of temporal-spatial context and physical constraints, and employ a manifold structure to connect high-dimensional text space with disentangled feature space. By introducing a self-correction database on the manifold, the system can continuously learn and dynamically adjust decision strategies. Experimental results demonstrate that our framework significantly outperforms existing LLM-based autonomous driving systems including GPT-Driver, DriveGPT4, and DriveLLaVA in collision rate metrics, validating its robustness and adaptability in complex environments.

The main contributions of this research include: (1) constructing a semantically enhanced autonomous driving temporal-spatial manifold enabling LLMs to maintain decision stability under anomalous inputs; (2) developing a test dataset containing multiple anomaly patterns, providing a benchmark for evaluating decision-making capabilities under anomalous perception inputs; (3) proposing a manifold-warping continuous learning mechanism, achieving resource-efficient knowledge accumulation; (4) experimentally verifying the framework's significant effect in reducing collision rates compared to diverse baseline methods, providing a new paradigm for enhancing L4 autonomous driving system safety.

## 2 RELATED WORK

### 2.1 FEW-SHOT OBJECT DETECTION IN AUTONOMOUS PERCEPTION

Few-shot learning in autonomous driving has largely focused on object detection. Approaches like Meta R-CNN Li et al. (2022b) use meta-learning to improve adaptation to new classes. Other methods enhance detection through various mechanisms, including feature re-weighting in FSRW Kang et al. (2019), attention in Attention-RPN Chen et al. (2024), graph convolutional networks in QA-FewDet Bulat et al. (2023), and metric learning in NP-RepMet Lu et al. (2022). While these techniques advance few-shot perception, the application of few-shot learning to decision-making systems remains largely underexplored.

### 2.2 LLM-BASED APPROACHES FOR AUTONOMOUS DRIVING

Large language models (LLMs) Guo et al. (2024); Zheng et al. (2023); Biderman et al. (2023) are being integrated into autonomous driving to leverage their pre-trained knowledge and few-shot capabilities. In planning and decision-making, frameworks like LanguageMPC Sha et al. (2023) and DriveGPT4 Xu et al. (2024) generate high-level, interpretable plans. For perception, models such as OccLLaMA Wei et al. (2024) and ContextVLM Sural et al. (2024) utilize multimodal fusion for robust scene understanding. LLMs also improve interpretability Zheng et al. (2024), enhance multi-agent collaboration Jiang et al. (2024), and aid in simulation Wang et al. (2024). Despite their promise, key challenges remain in achieving real-time performance, interpretability, and safety under anomalous inputs.

### 2.3 ANOMALY HANDLING IN AUTONOMOUS DRIVING SYSTEMS

Autonomous driving systems face three primary types of anomalies: perception uncertainties, prediction errors, and decision-making risks Yang et al. (2023a); Deng et al. (2021). These are traditionally addressed through methods like multi-sensor fusion Liu et al. (2021), probabilistic trajectory generation Huang et al. (2019); Li et al. (2022a), and incorporating uncertainty into planning via POMDP Duan et al. (2021). However, traditional anomaly detection is fundamentally limited in its ability to enumerate all possible error types. Instead of simply detecting and removing anomalies, our work argues for a new focus: ensuring correct system handling even when anomalies are present. This is particularly critical for few-shot learning, where models must maintain accurate inference despite sparse data and anomalous samples.

## 3 METHODOLOGY

### 3.1 AUTONOMOUS DRIVING SPACE CONSTRUCTION

We project textual driving scenario descriptions into a high-dimensional space, then map to a low-dimensional autonomous driving subspace through feature disentanglement. Our innovation uses spatiotemporal graphs as the fundamental representation, capturing both spatial relationships and temporal dynamics for enhanced robustness.

**High-dimensional Text Space Construction** The text input $T$ includes environmental obstacles and ego vehicle state:

$$\mathcal{O}_{info} = \{o_1, o_2, \ldots, o_n\} \tag{1}$$

Each obstacle $o_i$ is a triplet of type, position, and trajectory:

$$o_i = (t_i, \mathbf{c}_i, \mathcal{R}_i) \tag{2}$$

The ego vehicle state is represented as:

$$\text{VehicleState} = (\mathbf{c}, \mathbf{v}, \omega, a, \text{CanBus}, s_{\text{heading}}, \delta, H, G) \tag{3}$$

Complete input combines obstacles and vehicle state:

$$T = (\mathcal{O}_{\text{info}}, \text{VehicleState}) \tag{4}$$

Figure 2: The diagram shows the manifold construction process: high-dimensional text vectors undergo feature disentanglement, with autonomous driving aspects embedded into a pre-initialized Euclidean graph. This autonomous driving space is then concatenated with the original text vectors to form the complete manifold.

Text input is mapped to high-dimensional space:

$$\mathbf{h} = \text{Embed}(T) \in \mathbb{R}^d \tag{5}$$

**Spatiotemporal Graph-based Autonomous Driving Space Construction**   We construct $K$ spatiotemporal graphs for different scene aspects:

$$\mathcal{G}_k = (\mathcal{V}_k, \mathcal{E}_k^s, \mathcal{E}_k^t) \tag{6}$$

These graphs contain spatial edges $\mathcal{E}_k^s$ connecting entities at the same time point and temporal edges $\mathcal{E}_k^t$ connecting the same entity across time points.

Feature disentanglement extracts $K$ types of driving-related features:

$$\mathbf{h}_k = \mathbf{W}_k^\top \mathbf{h}, \quad k = 1, 2, \ldots, K \tag{7}$$

Each feature type enhances its corresponding graph:

$$\mathcal{G}_k' = (\mathcal{V}_k, \mathcal{E}_k^s, \mathcal{E}_k^t, \mathbf{h}_k) \tag{8}$$

Spatial edge weights represent entity relationships:

$$e_{k,i,j}^s = f_{\text{spatial}}(\mathbf{h}_k, v_{k,i}, v_{k,j}) \tag{9}$$

Temporal edge weights capture evolutionary relationships:

$$e_{k,i,j}^t = f_{\text{temporal}}(\mathbf{h}_k, v_{k,i}^t, v_{k,j}^{t+\Delta t}) \tag{10}$$

Finally, by fusing the $K$ feature graphs, we construct a unified autonomous driving space:

$$\mathcal{S}_{\text{AD}} = \text{Fusion}(\{\mathcal{G}_1', \mathcal{G}_2', \ldots, \mathcal{G}_K'\}) \tag{11}$$

This approach disentangles information into structured spatiotemporal representations while preserving semantics and physical constraints. The dual connection structure enables reasoning through temporal continuity and spatial consistency when facing anomalous inputs, enhancing system robustness.

## 3.2   Manifold Construction

We integrate spatiotemporal graphs with high-dimensional text embeddings into a rigorous manifold structure to represent traffic environments. This allows for dynamic adjustments via historical data, enhancing robustness against anomalous inputs.

### 3.2.1   Spatiotemporal Product Manifold

We model the manifold as a Riemannian product of its spatial and temporal components: $\mathcal{M} = \mathcal{M}_{\text{spatial}} \times \mathcal{M}_{\text{temporal}}$. This decomposition captures anomalies in both spatial relationships and temporal evolution.

A mapping function $\psi$ embeds each spatiotemporal graph $\mathcal{G}'_k$ onto the manifold as a point $p_k$:

$$p_k = \psi(\mathcal{G}'_k) = (\psi_s(\mathcal{E}^s_k), \psi_t(\mathcal{E}^t_k)) \tag{12}$$

The tangent space is represented by the direct sum of its components: $T_{p_k}\mathcal{M} \cong T_{p^s_k}\mathcal{M}_{\text{spatial}} \oplus T_{p^t_k}\mathcal{M}_{\text{temporal}}$. Large language models are then used to extract a low-dimensional representation from this high-dimensional space.

### 3.2.2 SELF-CORRECTING MANIFOLD WARPING

The core of our approach is the dynamic adjustment of the spatiotemporal manifold to handle anomalies. The process, outlined in Algorithm 1, leverages a **self-correction database** to identify and rectify problematic data points.

---

**Algorithm 1:** Concise Spatiotemporal Manifold Warping

**Data:** Manifold $\mathcal{M}$, graph $\mathcal{G}'$, database $\mathcal{D}$, thresholds $\epsilon_{s,t}$, learning rates $\eta_{s,t}$
**Result:** Warped manifold $\mathcal{M}'$
// Map the new graph to the manifold
$p = \psi(\mathcal{G}') = (\psi_s(\mathcal{E}^s), \psi_t(\mathcal{E}^t))$
// Check for anomalies using KL divergence
$D_{KL}(p|q^*) = D_{KL}(p_s|q^*_s) + D_{KL}(p_t|q^*_t)$
**if** $D_{KL}(p_s|q^*_s) > \epsilon_s$ **then**
$\quad$ $\mathcal{A}_s \leftarrow \mathcal{A}_s \cup \{p\}$ $\qquad\qquad\qquad$ // Identify spatial anomaly
$\quad$ $\Delta\mathbf{v}_s \leftarrow \text{Search}(\mathcal{D}, \text{spatial\_type})$ $\qquad$ // Retrieve correction
$\quad$ $\mathcal{M}'_s = \mathcal{M}_s + \Delta\mathbf{v}_s$ $\qquad\qquad\qquad$ // Apply spatial warp
**end**
**if** $D_{KL}(p_t|q^*_t) > \epsilon_t$ **then**
$\quad$ $\mathcal{A}_t \leftarrow \mathcal{A}_t \cup \{p\}$ $\qquad\qquad\qquad$ // Identify temporal anomaly
$\quad$ $\Delta\mathbf{v}_t \leftarrow \text{Search}(\mathcal{D}, \text{temporal\_type})$ $\qquad$ // Retrieve correction
$\quad$ $\mathcal{M}'_t = \mathcal{M}_t + \Delta\mathbf{v}_t$ $\qquad\qquad\qquad$ // Apply temporal warp
**end**
$\mathcal{M}' = \mathcal{M}'_s \times \mathcal{M}'_t$ $\qquad\qquad\qquad\qquad$ // Update manifold

---

We first introduce the "self-correction database" $\mathcal{D}$, which stores problematic points and related experiences:

$$\mathcal{D} = \{d_1, d_2, \ldots, d_L\} \tag{13}$$

Next, we construct a warping mapping $\tau = (\tau_s, \tau_t)$ to adjust the manifold's geometry, which decomposes into spatial and temporal components. This allows us to make separate adjustments for different anomaly types. We use the **Fisher information metric** to measure distribution differences with KL divergence:

$$D_{\text{KL}}(p|q) = D_{\text{KL}}(p_s|q_s) + D_{\text{KL}}(p_t|q_t) \tag{14}$$

By setting thresholds $\epsilon_s$ and $\epsilon_t$, we identify anomalous points and categorize them into spatial ($\mathcal{A}_s$) and temporal ($\mathcal{A}_t$) sets:

$$\mathcal{A}_s = \{p \in \mathcal{A} \mid D_{\text{KL}}(p_s|q^*_s) > \epsilon_s\} \tag{15}$$
$$\mathcal{A}_t = \{p \in \mathcal{A} \mid D_{\text{KL}}(p_t|q^*_t) > \epsilon_t\} \tag{16}$$

For each anomalous point, a correction vector is calculated through information geometry gradients:

$$\Delta\mathbf{v}_s = -\eta_s \nabla_{\theta_s} D_{\text{KL}}(p_s|q^*_s) \tag{17}$$

This gradient-based adjustment morphs the manifold towards more reliable decision regions, thereby enhancing decision-making capabilities when faced with anomalous inputs.

### 3.3 REASONING AND TEXT SPACE MAPPING

Based on our spatiotemporal product manifold, we describe how to perform reasoning and map results back to understandable text form, leveraging spatiotemporal decomposition to handle anomalous perception inputs.

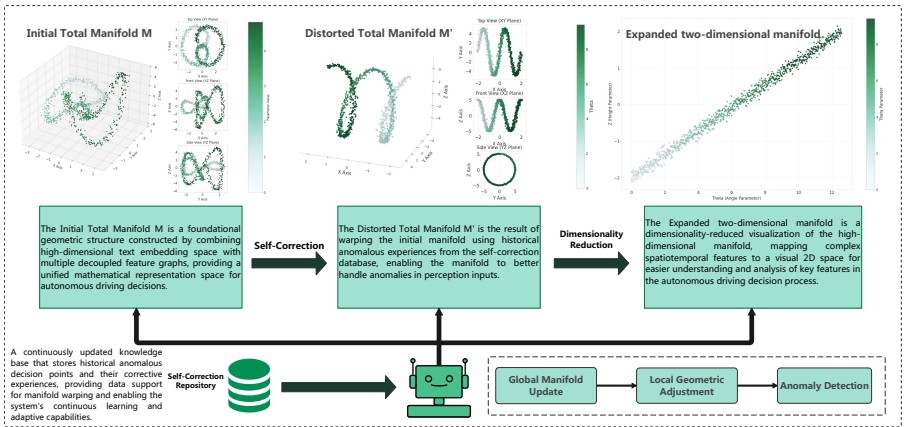

Figure 3: The diagram shows how the self-correction database warps the manifold. For each construction instance, similar scenario data is retrieved from the database and used to transform the manifold from an irregular shape into a more manageable configuration for further processing.

**Reasoning on Spatiotemporal Manifold** Our reasoning process predicts vehicle trajectory for the next three seconds on the manifold $\mathcal{M}'$, generating position information for six time steps. We define the trajectory prediction model as:

$$\mathcal{P} = \mathcal{L}_{\text{predict}} : \mathcal{M}' \times \mathcal{S} \to \mathbb{R}^{6 \times 2} \tag{18}$$

where $\mathcal{S}$ represents the current vehicle and environment state, and $\mathbb{R}^{6 \times 2}$ represents position information for six future time steps.

For input $T = (\mathcal{O}_{\text{info}}, \text{VehicleState})$, we obtain the embedding vector through feature disentanglement and spatiotemporal graph construction:

$$p = \psi(\mathcal{G}'_1, \mathcal{G}'_2, \ldots, \mathcal{G}'_K) = (p_s, p_t) \in \mathcal{M}'_s \times \mathcal{M}'_t \tag{19}$$

We then perform reasoning using the large language model:

$$\mathbf{R} = \mathcal{P}(p, \mathcal{S}) = (\mathbf{r}_1, \mathbf{r}_2, \ldots, \mathbf{r}_6) \in \mathbb{R}^{6 \times 2} \tag{20}$$

Each $\mathbf{r}_j = (x_j, y_j)$ represents the predicted position at the $j$-th future time step.

To evaluate reliability, we use Representation Engineering (RepE) to quantify confidence:

$$A_c = \{\text{Rep}(M, T_c(s_i))[-1] \mid s_i \in S\} \tag{21}$$

We calculate representation difference vectors:

$$\{A_c^{(i)} - A_c^{(j)}\} \tag{22}$$

The final confidence estimate is computed as:

$$\mathcal{C} = \text{Rep}(M, x)^T v \tag{23}$$

**Mapping to Text Space** We map inference results back to text space using the function $\Phi = \mathcal{L}_{\text{map}} : \mathbb{R}^{6 \times 2} \times \mathbb{R} \to \mathcal{H}'$. This mapping provides trajectory predictions and explains anomaly handling, making the decision process transparent and interpretable. The function consists of three components: $\Phi_{\text{proc}} = \mathcal{L}_{\text{describe\_process}}(\mathcal{P})$ generates a natural language description of the reasoning process; $\Phi_{\text{result}} = \mathcal{L}_{\text{describe\_result}}(\mathbf{R})$ converts the predicted trajectory to text; and $\Phi_{\text{conf}} = \mathcal{L}_{\text{describe\_confidence}}(\mathcal{C})$ expresses the confidence level in text. Together, these components form the complete textual description $\Phi(\mathbf{R}, \mathcal{C}) = \mathcal{L}_{\text{map}}(\mathbf{R}, \mathcal{C})$.

### 3.4 DETECTION AND CONTINUAL LEARNING

Based on the spatiotemporal product manifold framework, we design detection and continual learning methods to ensure system robustness and adaptability in complex environments.

**Construction of the Detection Module**   The detection module evaluates system performance through three complementary components:

**Logical Detection**: Verifies the model's consistent understanding of the spatiotemporal manifold. We design a question set $\mathcal{Q} = \{q_1, q_2, \ldots, q_M\}$ expressing the same logical problem from different perspectives. Consistency score:

$$\mathcal{L}_{\text{logic}} = \frac{1}{M} \sum_{m=1}^{M} \mathcal{I}\left(\mathcal{L}(q_m)\right) \tag{24}$$

Based on semantic similarity:

$$\mathcal{I}\left(\mathcal{L}(q_m)\right) = \text{Similar}\left(\mathbf{v}_{\mathcal{L}(q_m)}, \mathbf{v}_{\text{std}}\right) \tag{25}$$

**Confidence Space Detection**: Utilizes confidence score to quantify result reliability:

$$\mathcal{L}_{\text{conf}} = \mathcal{C} = \text{Rep}(M, x)^T v \tag{26}$$

**Prediction Detection**: Evaluates risk level of the generated trajectory:

$$\mathcal{R} = \mathcal{L}_{\text{risk}}(p_{\mathbf{R}}) \tag{27}$$

The comprehensive evaluation metric is the product of the three component scores:

$$\mathcal{D}_{\text{output}} = \prod_{i=1}^{3} \mathcal{D}_i \tag{28}$$

**Continual Learning on Spatiotemporal Manifold**   The continual learning mechanism uses real-time feedback to optimize system performance. The system judges result acceptability through an evaluation threshold:

$$\mathcal{D}_{\text{output}} < \Theta \quad \text{then trigger the continual learning mechanism} \tag{29}$$

When problems are detected, the system identifies and records problem points to the self-correction database:

$$\mathcal{D} = \mathcal{D} \cup \{d_{\text{new}}\} \tag{30}$$

Based on the updated database, the spatiotemporal manifold is rewarped:

$$\mathcal{M}'' = \tau(\mathcal{M}', \mathcal{D}) = (\tau_s(\mathcal{M}'_s, \mathcal{D}_s), \tau_t(\mathcal{M}'_t, \mathcal{D}_t)) \tag{31}$$

Trajectory prediction and evaluation are performed again on the new manifold, calculating the relative error:

$$\epsilon = \left| \frac{\mathcal{D}'_{\text{output}} - \mathcal{D}_{\text{output}}}{\mathcal{D}_{\text{output}}} \right| \tag{32}$$

The system decides whether to continue iterating or output results based on this error. Through this iterative optimization mechanism, the system can continuously learn and adapt when encountering new anomalies, improving robustness without requiring large-scale retraining.

# 4   EXPERIMENTS

## 4.1   DATASET

We developed a specialized anomalous data test set targeting autonomous driving decision modules, comprising 2,730 frames extracted from the nuScenes dataset and converted to bird's-eye view representations. The dataset systematically simulates perception anomalies induced by extreme weather conditions, cyber attacks, and sensor failures. These anomalies manifest as physically implausible approaches (obstacles with excessive speed or impossible directional changes), non-physical retreats (abnormal jumping or retreating behaviors), single-frame coordinate anomalies (sudden position disturbances), and distant vehicle sudden proximity events (remote obstacles abruptly appearing nearby). This comprehensive dataset effectively models real-world perception challenges such as sensor misjudgments in adverse weather, electromagnetic interference, data tampering through hacking, instantaneous sensor failures, and vehicle skidding on slippery surfaces, providing a robust benchmark for evaluating decision system resilience.

## 4.2 EXPERIMENTAL SETUP

All experiments were implemented based on the open-source LLama3:8b model, initialized and deployed through the Ollama framework. The computing platform was equipped with dual NVIDIA GeForce RTX 3090 GPUs (24GB VRAM/card) to support intensive inference computations. The experimental datasets included standard scenarios and self-constructed anomalous perception input scenarios, with the latter specifically designed to test system robustness under perceptual disturbances.

## 4.3 EXPERIMENTAL RESULTS

We conducted comprehensive comparative evaluations of the proposed LLM-ADF framework against mainstream autonomous driving decision-making methods on both UniAD and ST-P3 datasets. The experiments encompassed traditional rule-based methods, probabilistic decision methods (POMDP), model predictive control (MPC), deep reinforcement learning (DRL), Decision Transformer, as well as existing large language model approaches (DriveGPT4, GPT-Driver, DriveLLaVA).

**Experimental Setup**: Each method was tested under two conditions: normal perception inputs ("no error") and anomalous perception inputs ("add error"), evaluating collision rate performance across 1-second, 2-second, and 3-second time windows.

Table 1 presents performance comparisons between our framework and baseline methods under both UniAD and ST-P3 datasets. Our approach demonstrates consistent superiority across all experimental conditions and time horizons.

**Key Findings**:

**Overall Performance**: The proposed LLM-ADF method achieved the lowest collision rates across all test conditions. On the UniAD dataset, our method achieves average collision rates of 24.93% (no error) and 28.64% (add error), representing significant improvements over the best baseline DriveLLaVA (27.86% and 32.46% respectively). On ST-P3, the improvements are even more pronounced with 7.82% (no error) and 10.45% (add error) compared to DriveLLaVA's 9.78% and 13.00%.

Table 1: Baseline Comparison - Collision Rate (%)

| Dataset | Condition | Time | Rule-based | POMDP | MPC-based | DRL (SAC) | Decision Transformer | TransFuser++ | DriveGPT4 | GPT-Driver | DriveLLaVA | Ours (LLM-ADF) |
|---------|-----------|------|-----------|-------|-----------|-----------|---------------------|--------------|-----------|------------|------------|----------------|
| UniAD | no error | 1s | 12.45 | 9.87 | 8.23 | 6.89 | 6.12 | 5.78 | 5.89 | 5.64 | 5.23 | **4.67** |
| | | 2s | 28.73 | 24.16 | 21.67 | 19.34 | 17.89 | 16.92 | 17.23 | 18.05 | 16.45 | **14.23** |
| | | 3s | 89.56 | 78.45 | 75.34 | 68.92 | 65.23 | 62.45 | 64.78 | 72.18 | 61.89 | **55.89** |
| | add error | 1s | 15.82 | 12.34 | 10.95 | 9.12 | 8.34 | 7.91 | 8.15 | 8.65 | 7.78 | **6.12** |
| | | 2s | 34.91 | 29.58 | 26.43 | 23.78 | 22.15 | 21.34 | 21.89 | 21.43 | 20.67 | **17.45** |
| | | 3s | 92.84 | 83.72 | 80.91 | 74.56 | 71.67 | 69.78 | 71.34 | 74.44 | 68.92 | **62.34** |
| ST-P3 | no error | 1s | 8.92 | 6.15 | 5.78 | 4.23 | 3.89 | 3.45 | 3.67 | 4.70 | 3.12 | **2.45** |
| | | 2s | 19.45 | 14.73 | 13.24 | 10.67 | 9.23 | 8.67 | 9.12 | 9.77 | 8.34 | **6.78** |
| | | 3s | 41.67 | 32.89 | 29.45 | 21.89 | 19.56 | 18.23 | 19.45 | 23.93 | 17.89 | **14.23** |
| | add error | 1s | 11.78 | 8.91 | 7.82 | 6.45 | 5.67 | 5.23 | 5.34 | 7.33 | 4.89 | **3.67** |
| | | 2s | 24.32 | 18.45 | 16.89 | 13.92 | 12.45 | 11.78 | 12.23 | 12.78 | 11.45 | **9.23** |
| | | 3s | 47.83 | 38.67 | 34.78 | 26.78 | 24.33 | 23.45 | 24.67 | 26.69 | 22.67 | **18.45** |

**Robustness Advantage**: Under anomalous input conditions, LLM-ADF demonstrated the strongest robustness with only 14.9% performance degradation on UniAD and 33.6% on ST-P3, significantly lower than other large language model methods which typically show 17-21% degradation.

**Temporal Stability**: As the prediction time window increased, LLM-ADF exhibited the slowest performance degradation. The collision rate increases from 1s to 3s show our method maintains better control over longer prediction horizons compared to all baseline methods.

**Cross-dataset Generalization**: LLM-ADF maintained consistent relative advantages across datasets of different complexity levels. The method shows particularly strong performance on the more challenging ST-P3 dataset, validating its generalization capability.

**Comparative Analysis**: Traditional methods (Rule-based, POMDP, MPC-based) show significantly higher collision rates, with rule-based methods performing worst.

Deep learning approaches (DRL, Decision Transformer, TransFuser++) demonstrate improved performance but still lag behind LLM-based methods. Among LLM approaches, our LLM-ADF framework consistently outperforms existing methods including DriveGPT4, GPT-Driver, and DriveLLaVA.

The results indicate that the few-shot learning-based large language model autonomous driving framework has significant advantages in handling perceptual uncertainty and anomalous inputs, providing an effective solution for achieving safe and reliable autonomous driving decisions. These findings validate the effectiveness of our spatiotemporal reasoning approach and self-correction mechanisms in enhancing system robustness while preserving decision accuracy.

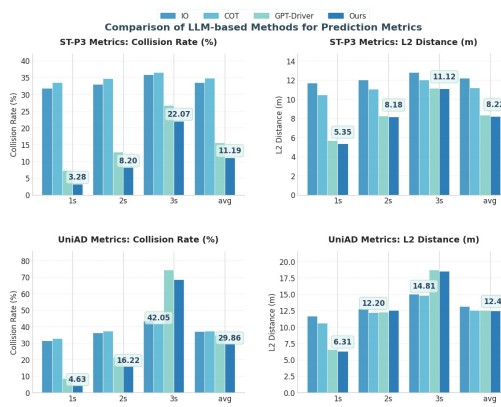

Figure 4: The result for the experiments.

## 5 DISCUSSION

Our framework significantly outperforms existing LLM-based methods including GPT-Driver, DriveGPT4, and DriveLLaVA in collision rate metrics under anomalous testing conditions through a compact, feature-disentangled autonomous driving semantic space. This approach enhances domain-specific semantic density, enabling LLMs to focus on critical information for deeper analysis. Theoretically grounded in information geometry, our method recognizes that anomalous inputs create distributional shifts in high-dimensional space; our spatiotemporal manifold mapping efficiently filters irrelevant noise while enhancing key features. The manifold's local geometric properties facilitate semantic consistency restoration through contextual reasoning despite anomalies.

We address continual learning challenges through an LLM-based self-assessment system that dynamically adjusts the statistical manifold's geometric structure via error instances stored in a self-correction database. This experience-based optimization enhances few-shot learning capabilities without requiring large-scale datasets. By leveraging pre-training knowledge with spatiotemporal decomposition, our system continuously optimizes in data-scarce environments.

From an industrial perspective, our solution enhances L4-level system robustness while enabling resource-efficient deployment. The semantic density enhancement methodology extends beyond LLMs to other model architectures, offering flexibility for various applications.

Future work will address current limitations, including expanding anomalous dataset diversity, optimizing the self-correction database's efficiency as it grows, and investigating the observed "vaccine effect" whereby certain anomalous data actually improves decision accuracy. This counterintuitive phenomenon merits deeper analysis in subsequent research.

## 6 CONCLUSION

This paper introduces a novel framework for handling anomalous perception inputs in autonomous driving through spatiotemporal decomposition of statistical manifolds. By leveraging LLMs' contextual understanding within a product manifold architecture, our approach separately addresses spatial and temporal anomalies. Experiments show collision rates reduced by up to 22.0% compared to GPT-Driver and achieving state-of-the-art performance among LLM-based systems while maintaining trajectory accuracy. Our self-correction mechanism enables continuous learning without extensive retraining—crucial for real-world deployment where anomalous data is rare but critical. Through enhanced semantic density and manifold-based reasoning, we bridge the gap between perception uncertainty and robust decision-making, advancing autonomous driving safety and establishing a foundation for resilient Level 4 systems capable of handling perceptual anomalies in complex environments.

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
