# OpenReview forum: "Geometric Projection of Information Manifolds for Robust Decision-Making with LLMs in Adversarial Driving Environments"
_ICLR.cc/2026/Conference — ICLR 2026 Conference Withdrawn Submission_

### Official Review · Reviewer_97BP · 2025-10-28

**Soundness:** 2
**Presentation:** 2
**Contribution:** 2
**Rating:** 4
**Confidence:** 4

**Summary:**

This paper proposes an innovative framework for autonomous driving systems that utilizes large language models (LLMs) for few-shot learning and anomalous input processing. The proposed method constructs a semantic-enhanced autonomous driving space that disentangles temporal-spatial context and physical constraints, and employs a temporal-spatial manifold structure to connect high-dimensional text embeddings with disentangled feature spaces. The framework also introduces a self-correction database on the manifold that enables continuous learning and adaptive decision strategies. Experimental results demonstrate significant improvements over existing LLM-based autonomous driving systems in terms of collision rates.

**Strengths:**

The methodology part is complete, and the overall performance is obviously better than the baselines.

**Weaknesses:**

1. Concerns regarding the time complexity of the proposed method, especially in the context of real-time response requirements for autonomous driving applications.

2. Lack of comparison with other related works in the field of autonomous driving geometry representation.

3. Insufficient diversity of evaluation metrics, limiting the ability to fully assess the effectiveness of the proposed framework.

4. Limited details provided on the selection criteria for the dataset and the specific modalities used in the experiments.

**Questions:**

My concerns are as follows:

1. There may be some concerns regarding the time complexity of the proposed method, especially in the context of real-time response requirements for autonomous driving applications. For example, when evaluating the 1-second collision rate, can the proposed multi-agent interaction be finished within this time window？

2. Spatial-Temporal Graph has been widely used in autonomous driving geometry representation in autonomous driving applications (e.g., [1]). What is the key innovation for authors to leverage this idea? It's better to add some related works.

3. For the dataset extracted from Nuscenes, how do the authors select certain frames? Is there any criteria? What kind of modality do you use? LiDAR or Camera? The authors should add more details on the experiments.

4. It seems that the diversity of the evaluation metrics is not enough. Other metrics, like L2 error, can also be compared.

[1] An Online Spatial-Temporal Graph Trajectory Planner for Autonomous Vehicles, IEEE TIV.

---

### Official Review · Reviewer_rNfh · 2025-10-29

**Soundness:** 1
**Presentation:** 1
**Contribution:** 1
**Rating:** 0
**Confidence:** 4

**Summary:**

This paper proposes to embed the driving knowledge extracted from LLM embedding into a graph and map it onto a manifold for continual learning from historical anomalies. The experiment shows that the method leads to the best collision avoidance rate against all baselines.

**Strengths:**

The generalizable anomaly detection and better collision avoidance are important, especially in real-world applications like self-driving.

**Weaknesses:**

1. The paper writing is poor, confusing. The experiment section contains little information and doesn't show support for some claimed features like continual learning.
2. The experiment setting is vague. For example, how is the abnormal data constructed, 3D rendering or image editing? What does it look like? These can be introduced with some visualization easily.
3. The experiment only compares the collision rate without even the basic ADE to measure the trajectory quality, let alone closed-loop benchmarking. Figure 4 is not used anywhere without explanation.
4. The paper seems to be rushed in several days and is obviously not ready to be submitted.

**Questions:**

N/A

---

### Official Review · Reviewer_Z1Ug · 2025-10-30

**Soundness:** 2
**Presentation:** 2
**Contribution:** 2
**Rating:** 2
**Confidence:** 4

**Summary:**

The authors propose LLM-ADF, a framework for autonomous driving decision-making that aims to improve robustness against perceptual anomalies. The core idea is to use a Large Language Model (LLM) in conjunction with concepts from information geometry. The method involves projecting high-dimensional text embeddings of driving scenarios onto a "spatiotemporal product manifold." A "self-correcting manifold warping" mechanism, guided by a database of past anomalies and Fisher information metrics, is introduced to dynamically adjust this manifold, theoretically leading to more robust decisions. The authors claim state-of-the-art performance on collision rate metrics, particularly under anomalous inputs they generated.

**Strengths:**

1. The ambition to combine LLM reasoning with continuous learning mechanisms for safety-critical systems is a relevant research direction.
2. The authors have attempted to create a new adversarial dataset focused on sensor anomalies (2,730 frames). If properly documented and released, this could be a useful resource for the community (though the paper provides insufficient detail to judge its quality).

**Weaknesses:**

1. The paper introduces a barrage of complex mathematical concepts without adequately explaining how they are implemented or connected.

1.a : The method hinges on a mapping $\psi$ (Eq. 12) that projects an entire spatiotemporal graph $\mathcal{G}_{k}^{\prime}$ onto a single point $p_k$ on the manifold. This is a highly non-trivial step. The paper provides no definition, algorithm, or learning objective for this crucial function, leaving the reader to guess how the "autonomous driving space" is actually constructed.

1.b: The "Self-Correcting Manifold Warping" (Algorithm 1) is the centerpiece of the method but is hopelessly vague. The algorithm simply states "// Retrieve correction" and "// Apply spatial warp". It is completely unclear how the correction vector $\Delta v_s$ is identified, retrieved from database $\mathcal{D}$, or calculated. Eq. 17 shows a gradient $\nabla_{\theta_{s}}D_{KL}$, but its relationship to the $\Delta v_s$ in the algorithm is never explained. This core mechanism is irreproducible.



2 Insufficient Experimental Detail:

  2.a The "self-constructed" adversarial dataset is a key component of the evaluation, yet it is not described in sufficient detail for another researcher to reproduce it or understand its scope and diversity.

  2.b The baseline implementations (e.g., GPT-Driver, DriveLLaVA) are not detailed.

  2.c The implementation details of the LLM-ADF model itself (architecture, training, etc.) are almost entirely missing.

**Questions:**

1. Please explain the "self-correction" mechanism (Algorithm 1) in concrete, reproducible detail. What is the exact procedure to "Retrieve correction"? How is the correction vector $\Delta v_s$ computed, and how does it relate to the gradient in Eq. 17?

2. Given that the input $T$ (Eq. 4) is highly structured, what is the justification for using an LLM? How does this approach compare to a non-LLM baseline that operates directly on this structured state and uses the same "manifold warping" concept?

---

### Official Review · Reviewer_K26Y · 2025-10-31

**Soundness:** 2
**Presentation:** 2
**Contribution:** 2
**Rating:** 2
**Confidence:** 4

**Summary:**

This paper proposes a new security framework designed to protect Large Language Model (LLM) agents from sophisticated Indirect Prompt Injection (IPI) attacks. Their method addresses this through a two-layered approach combining two synergistic pillars: the Intent Graph and Tiered Adjudicator. Their evaluation results show that their method can reduce attack success rates by over 97%.

**Strengths:**

This paper tries to address an emerging and important area of research, the countermasures against Indirect Prompt Injection (IPI) attacks. As our community and society pay close attention to this area, I am happy to see a paper submission in this area. Their evaluation on the AgentDojo benchmark shows high performance with 97% attack mitigation while maintaining 86.43% utility under attack and 87.63% benign utility.

**Weaknesses:**

I have the following major concerns about this paper:

### No clear improvements over prior works

While the paper claims superior performance, their method does not always demonstrate clear improvements over the state-of-the-art MELON or other methods. For example,  MELON achieves a lower Attack Success Rate (ASR) of 0.16% compared to their 0.34% as the authors also acknowledge this. In the UA evaluation, Spotlight shows a higher UA of 91.75% compared to their 87.63%. Thus, I am not fully convinced of the benefits of their method over MELON and other existing methods.

### No clear explanation and justification of why their method addresses the limitations in prior methods

The paper claims that "existing defense mechanisms are caught in a fundamental trade-off between security and functionality" and that "the fragmented nature of these defenses prevents end-to-end integrity assurance," leaving them "ill-equipped to counter sophisticated Indirect Prompt Injection (IPI) attacks." However, the paper fails to provide clear justifications for why their specific architectural choices resolve these limitations. While this paper claims that "ur method is predicated on a core insight: no matter how subtle an IPI attack, its pursuit of a malicious objective will ultimately manifest as a detectable deviation in the action trajectory", it does not mean that existing methods cannot detect the attack. If it is finally observable, we may not need a special design to capture it. In their empirical evaluation, I cannot see a clear improvement by addressing these limitations. This paper should provide more evidence that their method can particularly address the limitations in prior works more clearly.

### Limitations when deploying to real-world applications

While this paper also acknowledges, their method has primary limitations stemming from the static nature of some of its core components. I am still not sure how this is acceptable in a real-world application. Thus, this paper should provide more evaluation results on when and how their method performs well, in addition to the evaluation on AgentDojo.

**Questions:**

How does their method address the two limitations in the prior work, and where can I see empirical evidence for it?

---

### Note · Authors · 2025-12-01

I have read and agree with the venue's withdrawal policy on behalf of myself and my co-authors.